# A Novel Perspective on Neuronal Control of Anatomical Patterning, Remodeling, and Maintenance

**DOI:** 10.3390/ijms241713358

**Published:** 2023-08-29

**Authors:** Emilie Jones, Kelly A. McLaughlin

**Affiliations:** Department of Biology, Tufts University, 200 Boston Avenue, Suite 4700, Medford, MA 02155, USA; emilie.jones@tufts.edu

**Keywords:** nervous system, CNS, PNS, development, morphogenesis, patterning, remodeling, regeneration, homeostasis

## Abstract

While the nervous system may be best known as the sensory communication center of an organism, recent research has revealed a myriad of multifaceted roles for both the CNS and PNS from early development to adult regeneration and remodeling. These systems work to orchestrate tissue pattern formation during embryonic development and continue shaping pattering through transitional periods such as metamorphosis and growth. During periods of injury or wounding, the nervous system has also been shown to influence remodeling and wound healing. The neuronal mechanisms responsible for these events are largely conserved across species, suggesting this evidence may be important in understanding and resolving many human defects and diseases. By unraveling these diverse roles, this paper highlights the necessity of broadening our perspective on the nervous system beyond its conventional functions. A comprehensive understanding of the complex interactions and contributions of the nervous system throughout development and adulthood has the potential to revolutionize therapeutic strategies and open new avenues for regenerative medicine and tissue engineering. This review highlights an important role for the nervous system during the patterning and maintenance of complex tissues and provides a potential avenue for advancing biomedical applications.

## 1. Introduction

The development of metazoan, multicellular organisms, starts with a single cell which gives rise to a myriad of highly specialized cell types. Throughout the process, cells must coordinate the cell behaviors necessary to build highly complex tissues and organs in a deterministic manner. One of the most important of these early tissues is the neuroectoderm. In vertebrates, this ectodermally derived tissue gives rise to the neural tube (NT), which will form the central nervous system (CNS), and neural crest cells (NCCs), which will contribute to an array of structures including peripheral nerves [1,2,3]. Interestingly, not only are neuroectodermal-derived tissues critical for the creation of the nervous system, they also play important roles during tissue patterning throughout development, the replacement of missing structures during regeneration, and the remodeling of existing structures [1,2,4,5,6,7]. Abnormalities in neuronal development do not only affect the cognitive and communicative abilities of an organism but also the early development of birth defects such as cleft palates as well the organism’s postnatal ability to grow and repair adult body structures [8,9,10]. This review begins by providing a brief background describing early developmental processes that contribute to tissue patterning events and will summarize the breadth of research describing the nervous system’s contributions to the building, and even rebuilding, of embryonic structures as well as a potential role for this system in the resolution of developmental defects. 

## 2. How Do We Build an Organism? From Embryo to Juvenile to Adult Body Plans

### 2.1. Neural Ectoderm in Development

In early chordate development, the portion of the ectoderm known as the neural plate begins to fold and fuse to become the neural tube during primary neural tube formation [11]. Primary neural tube (NT) closures ensure proper brain formation, while secondary closures at the lower sacral and caudal levels help modulate the patterning of the spinal cord [11,12]. Proper closure of the NT is essential for avoiding patterning defects such as spina bifida and craniorachischisis in which the central nervous system is exposed to amniotic fluid, resulting in severe neuroepithelial degeneration [11,13,14,15,16,17]. In addition to the CNS, the neural ectoderm also gives rise to one of the most diverse cell populations within the developing embryo, the neural crest cells (NCCs). Delaminating from the dorsal portion of the neural tube, NCCs constitute a transient, multipotent cell population that is essential for the development of the peripheral nervous system (PNS) as well as contribute to a plethora of non-neuronal structures such as the craniofacial skeleton and components of the cardiovascular system [4,18,19,20,21]. These versatile cells depart from the NT through an epithelial to mesenchymal transition, allowing them to migrate and differentiate into their final cell fates across the embryo [19,22,23,24,25,26]. A more complete and in-depth history of neural crest lineage migration can be found in a recent review by the Bronner lab [27]. 

### 2.2. The CNS as a Regulator of Craniofacial and Anatomical Patterning

In vertebrates, the central nervous system of the developing embryo is comprised of the brain and spinal cord, which help to regulate the overall sensory–motor functioning of the organism. The brain is in control of information processing and behavior and physiologic maintenance, while the spinal cord acts as the main communication network between the brain and the rest of the body [28]. Beyond its well-studied role as the primary control center for the adult organism, several studies uncovered additional roles for the CNS in tissue patterning during development. For example, experiments using *Xenopus laevis* demonstrated that the removal of the brain during early development led to diminished muscle mass as well as disorganization of the somites [29]. Similarly, early removal of the salamander nervous system via NT excision resulted in severely decreased muscle mass during limb development [30]. In mammals, the disruption of early neural functioning resulted in abnormal olfactory pathway development [31,32]. 

One of the best examples of the brain’s influence during the embryonic patterning of non-neuronal tissues can be observed during the development of the face [8,33]. Starting during early development, the nervous system begins laying the groundwork for craniofacial patterning through the coordination of cranial neural crest cells [34]. Cranial neural crest cells, a subpopulation of NCCs, are derived from the anterior neural tube and migrate to build an array of tissue types within the face such as connective tissue, cranial nerves and cranial jaw cartilages [23,35,36,37,38]. The brain and the face develop very closely, both molecularly via cranial neural crest cells signaling as well as mechanically through physical contact, creating a linked developmental relationship [39,40,41,42,43,44]. Because the development of each of these structures depends on the proper patterning of the other, several researchers postulated that the brain and the face are morphologically integrated, as the interaction of their respective traits leads to covariance within the system [44,45,46,47,48]. Additional evidence for this integration was described in a detailed study of craniosynostosis, which is a condition where the infant’s skull plates fuse together before the brain is fully grown, resulting in an abnormal head shape and an overall reduction in brain size [49]. Prior to this study, it was assumed that brain defects were a direct response to the physical parameters of the head [50,51,52]. However, Motch Perrine and colleagues (2017) argue that the development of many key brain structures and functions occur prior to the fusion of the infant’s skull plates. Thus, the timing of patterning events reveals that simple head morphology is not the only contributing factor that results in brain malformations in this condition. Likewise, research has shown that in cases of craniosynostosis driven by mutations in fibroblast growth factor receptors, such as Apert or Crouzon syndromes, there are also signs of brain mispatterning prior to improper suture closures [49,53,54,55,56,57,58,59]. Consequently, it is likely that the defects observed in brain development are due to multiple factors including the early disruption of key developmental processes within the CNS such as migration and proliferation and then later from physical compression via skull fusion. 

In addition to craniosynostosis, a wide range of abnormal phenotypes can arise when the nervous system deviates from normal development. Orofacial defects, the most common in the US, occur in one in every 700 births per year on average [60]. Many of these defects are contributed to, if not primarily caused by, improper patterning of the nervous system during early development [61]. For example, the loss of VEGFa, a growth factor involved in neurogenesis, in neural crest cells results in cleft palates, which constitute one of these common orofacial defects [62]. Neuroepithelial apoptosis, resulting from *TCOF1* mutations, leads to Treacher–Collins syndrome, which is a congenital birth defect defined by underdeveloped craniofacial structures such as jaws and cheekbones [63]. Additionally, Sonic hedgehog signaling from the forebrain acts as an organizing cue for developing jaw morphology, resulting in hypoplasia of the skeleton or disorders such as holoprosencephaly [64,65,66]. 

### 2.3. The Peripheral Nervous System’s Role in Developmental Patterning

Although studies examining neural development frequently focus on brain formation, proper establishment of both the central and peripheral nervous systems is vital for the overall patterning of the embryo. Interactions of peripheral nerves with adjacent developing structures have been shown to impact normal development across numerous species, potentially acting as a conserved signaling center. For example, in humans, the development of the autonomic nervous system, the branch of the PNS responsible for involuntary physiologic maintenance, is necessary for proper respiratory and cardiac functioning [67,68,69,70]. Additionally, in children who experience obstetrical brachial plexus injury at birth, the peripheral nerve damage can result in limb dysplasia characterized by reduced limb lengths and muscle atrophy [71]. 

In other mammals such as mice, peripheral sensory nerves known as Schwann cells are necessary for the guidance of blood vessel branching and alignment of the arteries in the developing skin [72]. Similarly, nerve growth factor directed innervation within the mouse femur has also been shown to be essential for vascularization and ossification during limb development [73]. Work with mice salivary glands demonstrated a vital role for parasympathetic innervation in the maintenance of progenitor cells necessary for epithelial organogenesis, which is a process important not only in the development of the gland but also tissue regeneration through adulthood [74]. Additionally, both fetal and neonatal denervation of the sciatic nerve in rats results in an absence of skeletal muscle specification and, thus, the inability to produce distinct adult muscle fibers [75]. In newborn rabbits, ablation of facial nerves resulted in stunted skeletal development with reduced muscle functioning [76]. Facial nerve ablation of prepubertal rabbits was also shown to result in misdirected growth of the snout due to the biomechanical effects of facial paralysis [77]. 

This control system extends beyond mammalian species as well. For example, the regulation of paraxial mesoderm programed cell death in chicken somites via neurotrophins has been shown to be crucial for sclerotome or skeletal differentiation [78]. Interestingly, proper innervation of the wings and legs of chick embryos was also found to be necessary for skeletal growth with poorly innervated groups showing a reduction in overall size of about 20 percent [79]. Additionally, chicken embryos rely on the NT-derived signal, neurotrophin-3, for proper skin formation via epithelial-to-mesenchymal conversion of dermatome progenitors [80]. Fish such as *Danio rerio* also rely on this signaling center for key developmental processes. In zebrafish, the misregulation of myelinating cells in the central and peripheral nervous systems shows corresponding defects in craniofacial and melanocyte morphologies as well as swimming behaviors [81]. 

Furthermore, though more often studied in adult organisms, neurotransmitters such as serotonin and dopamine play key roles in early development and, when dysregulated via pharmacologic modulation, can lead to defects in craniofacial structures such as cartilage and eyes as well as muscle, pigment and vascular patterning across the organism [82,83,84,85]. Serotonin in particular has been shown to have an immense impact on patterning throughout development, as described in [86]. 

## 3. How Do We Maintain Proper Patterning of the Adult Body Plan? 

### 3.1. Neuronal Control of Organ System Function, Patterning, and Maintenance

Even after embryonic development, the nervous system’s role in maintaining patterning and physiologic functioning remains vital throughout the lifetime of an organism. As cellular and molecular turnover is occurring every day, the adult body relies on neural circuitry to support the growth and upkeep of tissues such as skin, muscle, and vasculature [87,88,89]. The skin is a human’s largest and most dynamic organ; its roles include protecting the internal organs from the environment, helping with water regulation, and acting as a sensory conduit [90]. Because the skin plays such an important role, it has a high cellular turnover rate to maintain homeostasis [91,92,93,94]. Epithelial maintenance is very closely tied to sympathetic nerve functioning; when this functioning is disrupted due to diseases such as diabetes or Parkinson’s, a breakdown of proper epidermal homeostasis is observed [95,96]. Interestingly, in diabetic patients, the skin has difficulty regulating water due to damaged sympathetic nerve fibers, which leads to hyperhidrosis as well as dehydrated skin [97,98,99]. In addition, within the organism, the epithelia of the intestines also requires enteric neural inputs to properly regulate cellular differentiation, nutrient absorption, and the movement of microbes within the gut [100,101]. 

Beyond the epithelia, another highly proliferative organ, the liver, relies on the hepatic nervous system for its regenerative abilities via parasympathetic modulation of the vagus nerve [102,103]. Although the specific mechanisms by which hepatic innervation allows for the regulation of liver homeostasis vary between species, the autonomic nervous system has been shown to be vital for both cellular regeneration and apoptosis [104,105,106,107]. Even though the enteric nervous system plays an important role in maintaining liver function, it can also have harmful effects on gut health when misregulated [108]. Abnormal serotonin signaling as well as enteric neuronal activation has been shown to exacerbate irritable bowel syndrome pathogenesis [109,110,111]. 

Muscles are another example of a tissue that requires a good deal of maintenance, as they undergo a tremendous amount of morphologic remodeling over the course of an organism’s lifetime [112,113,114]. The sympathetic neurons of the heart are vital in regulating cardiomyocyte size, which is an important factor in heart disease [115]. In addition to its role in epithelial maintenance, the enteric nervous system is also an important regulator of intestinal muscular contractions and dilations, which is a process necessary for nutritional absorption [116]. Sympathetic control of the vascular system via neuromediators such as norepinephrine is responsible for the vasoconstriction of vascular smooth muscles, allowing for blood pressure regulation during exercise [117,118,119,120]. During periods of hypoxia, where oxygen availability is limited, the peripheral nervous system modulates vasodilation to increase muscle blood flow improving oxygen delivery [121,122]. Skeletal muscle atrophy or wasting is a common phenotype associated with demyelinating diseases such as multiple sclerosis and Charcot–Marie–Tooth disease due to both disuse and denervation [123,124,125,126,127]. Skeletal muscle wasting has also been seen in patients following acute strokes, with evidence showing that neuromuscular electrical stimulation can prevent or reduce these pattern alterations [128,129,130]. In mice models, research has shown that inflammatory signaling from the CNS influences the anabolic or catabolic regulation of skeletal muscle via the hypothalamic–pituitary–adrenal (HPA) axis [131]. 

In addition to the examples described above, the misregulation of the CNS and PNS can lead to a breakdown of sensory systems. The neurodegeneration that characterizes Parkinson’s syndrome also affects the functioning of the eyes; in early stages of the disease, the impaired regulation of neurotransmitters and metabolism of monoamines leads to decreased tear fluids, inflammation of the eyelids, and retinal thinning [132,133]. In the mouth of rats, denervation of the gustatory nerves, the chorda tympani and lingual nerves, alters the size and number of taste bud cells [134]. Similarly, recent work in mice has shown that sonic hedgehog (Shh)-expressing gustatory neurons play an important role in taste bud differentiation [135,136]. This was observed in humans prescribed the Shh inhibiting drug, sonidegib, for cancer treatment and later confirmed in mice [137,138]. 

While in most of the aforementioned cases, innervation was important for the general, normal maintenance of the system, recent research suggests that altered innervation could also play a role in disease. For example, in patients with prostate cancer, hyperinnervation has been shown to increase tumorigenesis via sympathetic neonerves and even metastasis via parasympathetic nerves [139]. A similar phenomenon has been shown in chronic pancreatic inflammation and later malignancy [140,141,142]. Additionally, work in mouse models has contributed evidence for therapeutic effects of denervation on gastric tumorigenesis [143]. A more complete assessment of the nervous system’s role in cancer regulation has been reviewed by Saloman et al. [144]. 

### 3.2. Transformation toward an Adult Patterning 

For many animals, part of their development toward creating an adult body plan requires the reorganization of whole tissue structures. One of the best studied examples of this remodeling has been examined in metamorphic animals. Metamorphosis is a highly coordinated process by which neuroendocrine signaling allows for the breakdown and rebuilding of juvenile structures to prepare an organism for adult functioning. Throughout this daunting undertaking, both the CNS and the PNS play key roles in supporting and permitting these repatterning events. For example, in insects such as *Drosophila melanogaster*, the metathoracic innervation to the indirect flight muscles is necessary for the formation of the dorsoventral muscles and, to a lesser extent, the dorsal longitudinal muscles through myoblast generation during metamorphosis [145,146]. The ablation of the pupal stage motor neurons not only leads to muscular degeneration but also issues with adult eclosion [147]. The innervation of *Manduca sexta* muscles has also been shown to be vital in myoblast proliferation during metamorphosis via EcR-B1 upregulation [148]. 

Some species of sea slugs such as the *Phyllaplysia taylori* and *Phesilla sibogae* use ampullary neurons to interact with their environment to make decisions on the timing of metamorphic induction, often delaying the onset of metamorphosis indefinitely and only proceeding when their sensory system detects optimal environmental conditions [149,150]. When their apical sensory organ is damaged or ablated, these organisms fail to respond to chemical metamorphic cues [151]. The role of the apical ganglion’s role in gastropod sensory and metamorphic maintenance has been established [152]. Interestingly, in sea squirts, it has been suggested that papillary sensory neurons and rostral trunk epidermal neurons act as chemoreceptors and mechanoreceptors, respectively [153]. Thus, it seems likely that ascidians use these sensory mechanisms to interact with their environment to initiate or delay metamorphic events through neurotransmitter signaling [153,154,155,156,157]. Another sensory system found in *Portunus trituberculatus* (swimming crab) has been reported to play an important role for tissue patterning. In this animal, the eyestalk neurosecretory system is vital for proper metamorphic transitioning through endocrine regulation [158,159]. Eyestalk ablation affects metamorphosis in a time-dependent manner with early ablation resulting in transition inhibition and later ablations resulting in morphological changes such as small dorsal span and furcae [159]. 

The nervous system also plays a vital role in vertebrate metamorphosis, which is observed most commonly in organisms such as frogs. In these amphibians, as well as related vertebrates, this process is primarily controlled by the regulation of thyroid hormone (TH) activity via the hypothalamus and pituitary gland, which use neural networks and neuroendocrine pathways to communicate environmental cues to trigger metamorphosis [160]. The process of TH-mediated metamorphosis has been thoroughly reviewed elsewhere, allowing this review to focus on additional routes of metamorphic pattern control [161,162,163]. For example, previous research demonstrated that the proper innervation of limb buds during early metamorphosis is necessary for proper bone and muscle maturation in developing *Rana pipiens* and *Xenopus laevis* limbs [164,165]. In addition, tadpoles with severe brain or spinal cord defects are unable to reabsorb their tail tissue during metamorphosis [165]. 

Fish also undergo a form of metamorphosis as they develop from larval to juvenile stages. For instance, during flatfish metamorphosis, the most dramatic pattern shift occurs when the ears and eyes shift to one side of the face. To assure that the oculomotor and vestibular systems remain functioning after this drastic shift, the central nervous system undergoes unique restructuring where the secondary vestibular neurons terminate on vertical extraoculomotor and trochlear nuclei, which is a process not seen in other vertebrate systems [166]. Furthermore, the pigment of these fish is regulated by the central nervous system via synaptic regulation with disrupted functioning causing dyschromia during metamorphosis [167]. *Danio rerio* show similar neuronal control of pigment patterning as their melanophores, xanthophores and iridophores travel along different peripheral nerves to create their multicolored striated appearance [168,169,170]. 

Although humans do not go through what we would classically consider metamorphosis, we do undergo many developmental transitions where our body plans and functions shift to accommodate lifestyle needs—for instance, during the prenatal to perinatal shift as well as sexual maturation during puberty. Researchers have seen indications that there may be shared mechanisms between amphibian metamorphosis and human development driving these transitional changes, potentially suggesting numerous uncovered roles for the nervous system in human patterning [171,172]. For example, studies have shown that humans utilize thyroid hormones T3 and T4, important endocrine modulators in amphibian metamorphosis, during the pre to postnatal transition period [173,174,175]. Additionally, studies have found that a genetic loci involved in the metamorphic timing of *C. elegans* and *Xenopus laevis* is conserved in humans for the modulation of sexual maturation [176,177,178]. Evidence of these shared mechanisms suggests a closer tie between metamorphosis and human growth and maturation than previously considered, and it may be an important route of exploration in the future. 

## 4. Resolution of Abnormal Body Plans 

### 4.1. Regeneration 

In addition to the important roles CNS and PNS play during the creation and maintenance of complex structures, the nervous system is also a key player during the regeneration of damaged tissues. During wounding events, many organisms require innervation of the surrounding tissues to produce neurotrophic factors necessary for apical epithelial cap (AEC), and later blastema, formation and functioning [179,180]. In mammalian regeneration and dermal healing, nerve-derived mesenchymal cells dedifferentiate to play a vital role in blastema maintenance and later for the development of new bone and skin tissues [181]. One of the best examples of regeneration can be seen in planaria, which can rebuild over 99% of their tissue from as little as 1/279th of their bodies [182]. Flatworms appear to utilize central nervous system and gap junction instructional cues to pattern their anterior/posterior axes, often resulting in double-headed worms when disrupted [183,184]. The CNS also appears to play a role in planaria fission frequency via mechanosensory neuron patterning [185]. In addition, these organisms also require regulation of the neurotransmitter serotonin to properly regenerate eye structures [186].

Another classic model for regeneration is salamanders, which have remarkable regenerative abilities, especially in limb regrowth. For example, some amphibians can regenerate entire limbs after a wounding event, which can occur in large part due to proper innervation of the injury location [7,187]. In both newts and axolotls, protein gradients are modulated by axons in the injured limb to promote blastema formation and allow for regeneration [188,189,190]. Although axolotls are highly regenerative, repeated injury to the limb bud reduces the overall size of the regenerate potentially due to reduced innervation [191]. When fully denervated, axolotls completely lose their ability to form the vital blastema; this phenotype can be rescued by the introduction of exogenous Neuregulin-1, suggesting a role for NRG1/ErbB2 signaling in nerve-dependent regeneration [192]. In addition to limb regeneration, axolotls also require the peripheral nervous system for bone regeneration of the mandible [193]. Although adult frogs lack the regenerative abilities of some of their fellow amphibians, pre-metamorphic tadpoles have very comparable regenerative abilities to salamanders, which they maintain until metamorphic stages [194]. As long as there is proper innervation, tadpoles have been shown to repair their tails and optic nerves as well as limbs [195,196,197,198]. Interestingly, this effect is not simply blastema-specific; research on tadpole tail regeneration has shown that even spinal cord manipulations far anterior of the wound site resulted in severe disruptions in regeneration [199]. In addition, in post-metamorphic amphibians, the telencephalon has shown regenerative capabilities dependent on reconnection with the olfactory nerve [200]. In anuran froglets, limb regeneration also shows nerve-dependent restrictions [201,202]. Although anuran amphibians such as *Xenopus laevis* tend to lose their regenerative capacities in adulthood, they retain some abilities to initiate regeneration, which can be enhanced by hyperinnervation; when the forelimb is hyperinnervated, there is increased coordination of patterning marked by increased cartilage branching and cellular proliferation [203]. 

Outside of amphibians, zebrafish also possess remarkable regenerative abilities similar to pre-metamorphic tadpoles and utilize facets of the nervous system to help coordinate cellular behaviors. For example, adult zebrafish utilize cardiac innervation for cardiomyocyte regeneration after heart injury via cholinergic signaling [10]. In addition to heart regeneration, zebrafish can also repair amputated fins as long as their nerve fibers are intact [204,205]. 

Although not generally thought of as regenerative, mammals also have some limited capacities for nerve-dependent repair. Adults who lose the tip of their finger are able to use Schwann cell precursors to form a blastema and thus regenerate correctly patterned digit tissue [206]. In denervated mouse digits, the regenerated tissue showed disorganization of bone as well as loose mesenchymal tissue [207]. Like zebrafish, mice were also shown to have cardiac regenerative abilities within their first week post-birth, assuming there is sufficient neural activity [10]. With the breadth of research supporting the nervous systems’ role in wound and injury repair, it may suggest a potential additional role in pathological tissue remodeling or the remodeling in response to injury or disease. 

### 4.2. Non-Metamorphic Remodeling 

Although there is a tremendous amount of research supporting the nervous system’s vast role in tissue restoration via regeneration, how we remodel existing, but abnormal, structures has been less well studied. Throughout an organism’s lifetime, they experience numerous challenges including injury, environmental changes, or birth defects, some of which are able to trigger remodeling events [208,209]. How the nervous system may influence or even drive these events is largely unexplored. However, a few previous studies provide evidence for a potential organizational role for neuronal tissues during non-metamorphic remodeling events. For example, adult snapping shrimp experience not only the regeneration of claws after removal but also subsequent axon-mediated remodeling of minor to major claws, allowing them to better survive in their environment [210,211]. Similarly, peripheral nerve patterning influences the size and shape of annually regrown deer antlers and, in cases of injury, can result in remodeling toward a unique tine pattern not seen in the original body plan [212]. Mollusks are another group that relies on neural regulation for patterning both in growth and remodeling. The shells of these organisms display unique, highly coordinated pigment patterns controlled by underling neural networks; during their lifespan, they maintain their patterning even during periods of growth or damage, using their complex network of axons to return to uniform pigment bands [213]. In fish such as the zebrafish and crucian carp, gills undergo remodeling events in response to hypoxia, likely initiated by neuroepithelial cells, to increase surface area and develop protruding lamellae [214,215]. After injury events in mammals such as mice and humans, neuropeptide Y, a peptide expressed by the CNS and PNS necessary for homeostatic maintenance, plays a vital role in both bone remodeling as well as cardiac remodeling via increased angiogenesis [216,217,218]. The necessity of proper innervation becomes increasingly apparent in individuals with nerve damage such as diabetic neuropathy, where wound healing is significantly delayed and often marked by increased scarring [9,219,220,221,222]. 

## 5. Conclusions and Future Perspectives

All across the animal kingdom, the nervous system is working to build, repair and maintain morphological patterning. Although the primary role for this system may be sensory integration and signaling, it is impossible to ignore the importance of neural control on form and function. The literature cited in this review highlights the intricate and versatile nature of the nervous system by demonstrating the nervous system’s vital secondary role in shaping and supporting patterning across not only species but also life stages. Thus, it stands to reason that this system may function as a fascinating new target for future therapeutic treatment research. The breadth of research described in this review strongly suggests that the nervous system may act as a vital signaling or communication center responsible for organizing and coordinating directional information to influence pattern formation throughout all stages of an organisms’ lifetime. Comparing these inter-species phenomena may be invaluable in deciphering underlying mechanisms that could be harnessed to mitigate or treat a wide range of developmental disorders. For example, work across regenerative species such as salamanders or zebrafish can better inform us about the potential role of mechanisms like hyperinnervation on appendage regrowth. These findings fill essential gaps in our understanding of human wound healing and provide new avenues for work in regenerative medicine. Research investigating the effects of damaged or mispatterned nerves on tissue development and remodeling could also be groundbreaking in preventing disorders such as muscle wasting, vastly improving the quality of life for many patients. Additionally, information on how the early nervous system influences developmental pattern formation may allow us to predict comorbidity between many common birth defects, potentially providing earlier routes for diagnosis and repair. By further investigating the functioning of neural components across developmental phases, we can hopefully uncover a potentially groundbreaking puzzle piece revealing novel routes to defect resolutions or even prevention.

## Data Availability

Not applicable.

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
