# Peer review of "A Novel Perspective on Neuronal Control of Anatomical Patterning, Remodeling, and Maintenance"

_ijms, 2023, doi:10.3390/ijms241713358_

Round 1

Reviewer 1 Report

In the manuscript sent for review, the Authors described a novel perspective on the control of neural anatomy. I find the topic of the manuscript interesting, and the whole work is well thought out. The Authors put a lot of work into the preparation of this interesting work, primarily referring to 213 publications, many of them are quite old, but a large part are recent reports.

My comments:

1. the first reference is in bold. Why?

2. the Authors in the introduction described the purpose of the work.. "as a potential role for this system in the 

resolution of developmental defects". Do you know what developmental defects? Did the Authors find this information in the available literature when preparing this manuscript?

Author Response

Response to Reviewer 1 Comments

Point 1. In the manuscript sent for review, the Authors described a novel perspective on the control of neural anatomy. I find the topic of the manuscript interesting, and the whole work is well thought out. The Authors put a lot of work into the preparation of this interesting work, primarily referring to 213 publications, many of them are quite old, but a large part are recent reports.

We appreciate your review of our manuscript and are delighted that you found this topic interesting.

Point 2. the first reference is in bold. Why?

Thank you for pointing this out. It was a formatting error that has been corrected in the revised document.

Point 3. the Authors in the introduction described the purpose of the work.. "as a potential role for this system in the resolution of developmental defects". Do you know what developmental defects? Did the Authors find this information in the available literature when preparing this manuscript?

Thank you so much for this suggestion. We have revised the introduction to include developmental defects most often associated with neuronal mispatterning. These examples are described in more detail in sections 2.2 as well as 2.3.

Reviewer 2 Report

From the title, the authors tried to cover the area of neural control of regeneration and remodeling of target-organs. However, this topic is so broad that a whole book could be devoted to it, rather than a small article. In general, the review looks very superficial, rather as a collection of separate facts, little connected with each other. There is no system in the presentation, data on invertebrates are listed with vertebrates, data from higher mammals - with lower ones without a clear hierarchy and sequence. To date, many signaling molecules, pathways and transcription factors are identified but authors mentioned only some of them. In the section on neurotransmitters, authors mentioned only serotonin and dopamine. However, many neuropeptides are released from autonomic synapses on the periphery and have trophic action, for example, neuropeptide Y. The same is for gases (NO, CO ets).

There are some wrong phrases, for example line 184. “Sympathetic control of neurohormones such as norepinephrine”. The first half of phrase is incorrect since the sympathetic neurons control their target-organs, not their own mediators. The second: norepinephrine is a classical neuromediator. Neurohormones traditionally include neuropeptides of the hypothalamus.

Author Response

Response to Reviewer 2 Comments

Point 1. From the title, the authors tried to cover the area of neural control of regeneration and remodeling of target-organs. However, this topic is so broad that a whole book could be devoted to it, rather than a small article. In general, the review looks very superficial, rather as a collection of separate facts, little connected with each other.

We appreciate your perspective on our work and completely agree that this is an important topic that could be easily expanded in a multitude of ways. Due to size limitations, this review is not intended to be a comprehensive analysis of regeneration and remodeling, but to rather to spark interest for people in related fields and highlight less-studied roles for the nervous system. We have cited numerous review papers within each subtopic to allow the reader to explore each process more in-depth if they so choose, allowing us to focus on the more novel aspects of nervous system dependent patterning.

Point 2. There is no system in the presentation, data on invertebrates are listed with vertebrates, data from higher mammals - with lower ones without a clear hierarchy and sequence.

We recognize that there are many different ways to organize the presentation of this research, especially considering it is a phenomenon which plays such a complex role across species. To highlight the importance of the nervous system’s role in patterning and tissue homeostasis across species, this review was organized by specific biological events that occur in diverse organisms.

Point 3. To date, many signaling molecules, pathways and transcription factors are identified but authors mentioned only some of them. In the section on neurotransmitters, authors mentioned only serotonin and dopamine. However, many neuropeptides are released from autonomic synapses on the periphery and have trophic action, for example, neuropeptide Y. The same is for gases (NO, CO ets).

We agree that there are numerous molecules and factors we could have included in this review.  However due to size restrictions, we unable to include a complete list of signaling pathways that play a role within these developmental processes. For interested readers, we included citations of many excellent review papers on this topic.  Similarly, the work on neurotransmitters is simply too vast to fit within the scope of this review but we mentioned serotonin and dopamine as potential starting points if the readers wanted to investigate that route.

Your comment on neuropeptide Y led us to read some fascinating articles on the neuropeptide that we have now included in the revised manuscript in lines 361-364.

Point 4. There are some wrong phrases, for example line 184 (187 in the revised document). “Sympathetic control of neurohormones such as norepinephrine”. The first half of phrase is incorrect since the sympathetic neurons control their target-organs, not their own mediators. The second: norepinephrine is a classical neuromediator. Neurohormones traditionally include neuropeptides of the hypothalamus.

Thank you for bringing this to our attention; we can see how our phrasing was confusing. We have corrected the text in the revised manuscript (line 187).

Reviewer 3 Report

We read with interest the perspective  article by Jones and McLaughlin where they provided a nice overview of the role of the CNS/PNS role in injury/wounding conditions and how they impact regeneration which highlights the potential of regenerative medicine and tissue engineering.

there are a few minor comments that will benefit this work,

It would be highly beneficial if the authors can describe the roles of different neural cellular components ad their contribution to adult stages of regeneration.

Second, the authors discussed the notion of injury as a condition for the regenerative role of the CNS./PNS; the readers would be interested if the authors highlight these mechanisms in examples of injury conditions such as stroke and brain trauma if this is feasible

Author Response

Response to Reviewer 3 Comments

We read with interest the perspective  article by Jones and McLaughlin where they provided a nice overview of the role of the CNS/PNS role in injury/wounding conditions and how they impact regeneration which highlights the potential of regenerative medicine and tissue engineering.

there are a few minor comments that will benefit this work,

Point 1. It would be highly beneficial if the authors can describe the roles of different neural cellular components ad their contribution to adult stages of regeneration.

Thank you so much for the feedback. We agree with your suggestion that a brief overview of the general neuronal mechanisms would be helpful to the reader, so we have included a few additional sentences and citations at the start of section 4.1 to better prepare the readers for the in-depth primary literature discussed in this review. The revised sections can be found in lines 287-292.

Point 2. Second, the authors discussed the notion of injury as a condition for the regenerative role of the CNS./PNS; the readers would be interested if the authors highlight these mechanisms in examples of injury conditions such as stroke and brain trauma if this is feasible

We appreciate this interesting idea and agree that examining the potential mechanisms of how the nervous system might interact during post CNS damage would be fascinating.  Unfortunately, due to page restrictions we are unable to expand our review to include a comprehensive discussion of this topic, but instead - included a few additional citations on the resulting morphological effects of CNS injuries (lines 192-196). Additionally, it is our hope that this review will spark new interest in this topic resulting in more research on the possible effects of stroke or brain trauma on regeneration and healing in humans.

Round 2

Reviewer 2 Report

The authors formally did some minor changes in their article. However, I stick to my opinion that the manuscript could not be published in the present form and it gives no useful information to the reader. The article should be totally rewritten. The theme is too broad and authors should increase the volume in several times at least to cover such wide area. I suggest to focus on a narrower topic, for example on nervous control of regeneration or similar.